# A Modular Smart Ocean Observatory for Development of Sensors, Underwater Communication and Surveillance of Environmental Parameters

**DOI:** 10.3390/s24206530

**Published:** 2024-10-10

**Authors:** Øivind Bergh, Jean-Baptiste Danre, Kjetil Stensland, Keila Lima, Ngoc-Thanh Nguyen, Rogardt Heldal, Lars-Michael Kristensen, Tosin Daniel Oyetoyan, Inger Graves, Camilla Sætre, Astrid Marie Skålvik, Beatrice Tomasi, Bård Henriksen, Marie Bueie Holstad, Paul van Walree, Edmary Altamiranda, Erik Bjerke, Thor Storm Husøy, Ingvar Henne, Henning Wehde, Jan Erik Stiansen

**Affiliations:** 1Institute of Marine Research, P.O. Box 1870 Nordnes, 5817 Bergen, Norway; jean-baptiste.danre@hi.no (J.-B.D.); henning.wehde@hi.no (H.W.); jan.erik.stiansen@hi.no (J.E.S.); 2Institute of Marine Research, Austevoll Research Station, 5392 Storebø, Norway; kjetil.stensland@hi.no; 3Department of Computer Science, Electrical Engineering and Mathematical Sciences, Western Norway University of Applied Sciences, 5063 Bergen, Norway; keila.lima@hvl.no (K.L.); ngoc.thanh.nguyen@hvl.no (N.-T.N.); rogardt.heidal@hvl.no (R.H.); lars.michael.kristensen@hvl.no (L.-M.K.); tosin.daniel.oyetoyan@hvl.no (T.D.O.); 4Aanderaa Data Instruments A/S, P.O. Box 103 Midtun, 5843 Bergen, Norway; inger.graves@xylem.com; 5Department of Physics and Technology, University of Bergen, 5007 Bergen, Norway; camilla.satre@uib.no (C.S.); astrid.skalvik@uib.no (A.M.S.); ingvar.henne@uib.no (I.H.); 6NORCE, Postboks 22 Nygårdstangen, 5838 Bergen, Norway; beatrice.tomasi@norceresearch.no (B.T.); bahe@norceresearch.no (B.H.); maho@norceresearch.no (M.B.H.); 7Norwegian Defence Research Establishment, 3191 Horten, Norway; paul.vanwalree@ffi.no; 8Technology Department, Aker BP ASA, 4020 Stavanger, Norway; edmary.altamiranda@akerbp.com; 9Kongsberg Discovery AS, P.O. Box 111, 3191 Horten, Norway; erik.bjerke@kd.kongsberg.com (E.B.); thor.husoy@kd.kongsberg.com (T.S.H.)

**Keywords:** marine sensors, acoustic communication, Internet of Underwater Things

## Abstract

The rapid growth of marine industries has emphasized the focus on environmental impacts for all industries, as well as the influence of key environmental parameters on, for instance, offshore wind or aquaculture performance, animal welfare and structural integrity of different constructions. Development of automatized sensors together with efficient communication and information systems will enhance surveillance and monitoring of environmental processes and impact. We have developed a modular Smart Ocean observatory, in this case connected to a large-scale marine aquaculture research facility. The first sensor rigs have been operational since May 2022, transmitting environmental data in near real-time. Key components are Acoustic Doppler Current Profilers (ADCPs) for measuring directional wave and current parameters, and CTDs for redundant measurement of depth, temperature, conductivity and oxygen. Communication is through 4G network or cable. However, a key purpose of the observatory is also to facilitate experiments with acoustic wireless underwater communication, which are ongoing. The aim is to expand the system(s) with demersal independent sensor nodes communicating through an “Internet of Underwater Things (IoUT)”, covering larger areas in the coastal zone, as well as open waters, of benefit to all ocean industries. The observatory also hosts experiments for sensor development, biofouling control and strategies for sensor self-validation and diagnostics. The close interactions between the experiments and the infrastructure development allow a holistic approach towards environmental monitoring across sectors and industries, plus to reduce the carbon footprint of ocean observation. This work is intended to lay a basis for sophisticated use of smart sensors with communication systems in long-term autonomous operation in remote as well as nearshore locations.

## 1. Introduction

Ocean observations in both shallow and deep waters are essential to (i) monitor environmental changes, (ii) ensure sound ocean management and (iii) enable sustainable ocean operation of marine industries. Observations by scientific vessels have a long history [1]. However, to increase the spatial and temporal resolution, there is a strong tendency to increase the number of observation platforms by including various concepts of automated platforms, for instance the free-floating buoys in the ARGO networks [1] or stationary buoys. The development in recent years by smart sensor technology is also on its way into oceanic and coastal environments.

European research strategy states: “Strengthening observation and monitoring capacities through enabling technologies, new platforms and sensors; addressing under-sampling, and ensuring that new environmental parameters can be rapidly and accurately measured” [2]. Furthermore, the Internet of Underwater Things (IoUT), a marine version of the Internet of Things (IoT, [3]) addresses these topics of automation, low power, low cost and smart sensors [4]. Key enabling elements for IoUT are automated and smart devices connected through wireless communication and a common data acquisition and data handling platform, enabling data fusion and machine learning.

One industry that both impacts and is highly dependent on the marine environment is aquaculture. The need for improved monitoring is therefore obvious. It should be emphasized, however, that the development described in this paper is more aimed at marine industries and marine environment surveillance in general rather than this specific industry. There are analogous needs for monitoring and surveillance in a range of marine industries, as well as in highly diverse marine environments.

Whereas seafood from wild-caught fisheries have been stagnant for several years, aquaculture is growing globally at a fast rate [5]. Global production from aquaculture was approximately 80 million tonnes in 2020 excluding algae [6]. This is about equivalent to the quantity of wild-caught fish. When algae are included, the production was approximately 125 million tonnes [6]. In global terms, Asian countries are leading in total production and in most sectors of aquaculture. By far, the largest producer is China, whereas in Europe, Norway leads the development of aquaculture and is also the world’s largest producer of marine finfish [6]. This has not been achieved without environmental concerns [7] as well as increasing spatial conflicts [8]. Access to oceanographic as well as biological data may enhance optimization of aquaculture operations, not the least with respect to biosafety. In this context, environmental monitoring across sectors could add to the access of relevant data to the different industries. Sensors operating from offshore wind or oil/gas facilities may provide useful data for aquaculture and vice versa.

In the present study, we chose to use aquaculture as the primary case, motivated by the access to a large-scale research facility for marine ecology and aquaculture but also well suited for hosting experiments with different sensors and transmission systems.

The technology is, however, equally relevant for other marine use cases, such as offshore oil and gas industry, offshore wind, offshore mining, as well as oceanographic monitoring for environmental modelling and research.

The following sections (Section 2, Section 3, Section 4, Section 5, Section 6, Section 7, Section 8, Section 9 and Section 10) describe both the methods, equipment and the preliminary results from the experiments with sensors, underwater communication and data treatment. Section 11 is a discussion on the concepts tried so far, and on the concept of a marine large-scale facility for testing sensors and wireless underwater communication for use across sectors.

## 2. Sensors

Key sensors deployed at the aquaculture research station are Doppler Current Profilers (Seaguard II by Aanderaa, Xylem, Aanderaa Instruments AS, Nesttun, Norway) for measuring directional wave and current parameters, temperature, pressure, dissolved oxygen and conductivity, as well as Seabird SBE-27-SMP (Seabird Scientific, Bellevue, WA, USA) CTDs for redundant measurement of temperature, conductivity and oxygen. The surface units contain batteries, a 4G modem and floating units (Figure 1).

The positions in the water-column by the Seaguard II and Seabird in the two different rigs are shown in Figure 2. The geographical position is given in Figure 3, and the bathygraphy of the area is shown in Figure 4. The positions are adjustable according to the experimental setup requirements. The data are transferred to the SFI Smart Ocean Data Exploration portal [9] (Figure 5). Data transmission and processing are explained below.

In addition to the stationary sensors, shorter surveys are performed with hydrophone and CTD casts from a vessel. It is the intention that more rigs will be added to the observatory and used, some on a permanent basis. Temporary rigs will also be used, in particular to test wireless communication over longer distances in the hydrographically complex fjord environment.

## 3. Communication

Communication via 4G network or cable is included. However, a key purpose of the observatory is to facilitate experiments with acoustic wireless underwater communications. Since 2022, short- and long-term performance tests of commercial and software-defined acoustic modems have been performed. Moreover, dedicated acoustic measurements have been performed to characterize the propagation channel and the background noise.

## 4. Background Noise

The transmission range of an acoustic modem is governed by its source level, which is a measure of the acoustic power, the propagation loss, and the ambient noise level.

Figure 6 shows noise spectra measured in November 2023, obtained with a hydrophone suspended at a depth of 25 m from the fish cage platform. The spectra are averaged over 60 s of data, excluding modem transmissions and noise of passing boats. The blue, red, and purple spectra are essentially the same between 3 and 35 kHz. This is the noise background that is always present, and includes a number of sharp peaks (tonals). Between 35 and 55 kHz there are strong, intermittent sources of sound, which have been identified as echo sounders used by the CRIMAC project [10], a project led by the Institute of Marine Research, also performing work at the Austevoll facility.

The green and yellow spectra reveal an elevated noise level at all frequencies. A possible cause is the pumping system that takes in water for the on-land Austevoll facilities. There are many noise sources at the Smart Ocean observatory, and the overall noise level is high compared with an open ocean environment.

## 5. Experimental Tests of Oceanographic Sensor Data Transmitted via Acoustic Link

A Kongsberg Discovery cNODE Modem MiniS with the Aanderaa Seaguard II Data Processing Unit (DPU) with CTD, ADCP and oxygen sensors was integrated. The DPU is also cable-connected to the 4G modem embedded in the surface buoy.

The objective of this integration was to test the underwater acoustic link between the North rig and the dock and to obtain a first experience on the system integration. An overnight test on the bench in the air was performed, with the minimum power level for the acoustic modems, to test the correct integration of the Seaguard II with the cNODE Modem MiniS [11]. Only 2 of 5933 packets were lost. This validated the integration, and we then deployed this Seaguard II and the cNODE Modem MiniS into the North rig and sent data packets containing CTD data only to the node deployed at the dock. Seaguard II was installed 21 m below the surface and the cNODE Modem MiniS was installed 24 m below the surface. The second cNODE Modem MiniS was installed in the dock close by the fish farm at a depth of 5 m. The two acoustic modems were at approximately 300 m from each other. CTD data were sent every 10 s in order to collect a first evaluation of the data link quality in the facility.

Packets were detected and decoded in real-time. The follow-up development is to integrate the cNODE Modem MiniS at both Seaguard II DPUs at the North Rig and at the South Rig and connect another cNODE Modem MiniS to the fish farm infrastructure. Thanks to the 4G connectivity with the 2 rigs and the ethernet connectivity at the fish farm, it will be possible to remotely change the parameters for the three nodes and evaluate the network throughput that can be achieved in this star topology with unicast asynchronous transmissions from each of the rig’s nodes to the receiving node at the fish farm. The information traffic will be asymmetrical with most of the data from the sensors to the sink node at the fish farm, thus posing the challenge of power management of the battery powered acoustic modem at the rigs.

Therefore, this development will need the integration of a power switch between the Seaguard II and the cNODE Modem MiniS so that this latter can be turned off during inactive periods. Synchronization for the wake-up time at the transmitter and the receiver will be performed through the 4G network, as the synchronization issue is left for a later development. In addition, the Seaguard II will encode the data into a binary format to reduce the packet length used by the underwater acoustic communication system. This work has allowed us to better understand the challenges when implementing the integration between a DPU with multiple sensor payloads and an underwater acoustic modem. Even if both systems have been extensively tested on their own, power management and data source encoding are key aspects to solve when designing the integrated system.

## 6. NORCE Software Defined Modem (SDM) and cNODE Modem MiniS

The primary objective of SFI Smart Ocean is the advancement of a smart and wireless underwater sensor network, encompassing all network layers spanning from the physical layer to the application layer. To enhance and expedite both development and research endeavours, a novel and versatile software-defined modem has been devised.

The NORCE SDM has been assembled primarily using easily accessible components. Core elements include a Linux computer, a sound card, an amplifier, and a transducer. The main part of the SDM is the Raspberry Pi4 computer, which manages not only the physical layer processing but also the link layer protocols. This configuration gives flexibility and adaptability to accommodate a spectrum of underwater communication protocols.

Ethernet POE+ is accessible at multiple points within the Austevoll fish farm. Leveraging this infrastructure, an ethernet connection from the SDM ensures the provision of both power and communication. This configuration facilitates remote access to all SDMs, enabling continuous monitoring and configuration adjustments. The capability to download and test new modulation schemes or link layer protocols seamlessly from any location enhances the adaptability of the system. Moreover, the setup facilitates straightforward monitoring of link quality, as packages can be transmitted acoustically and over ethernet, allowing for convenient receiver-based comparisons.

The potential placement of SDMs both inside and around the fish farm provides a valuable opportunity to explore communication within noisy environments characterized by numerous interfering infrastructures. In our initial investigations, we conducted channel sounding experiments spanning the fish farm, revealing a myriad of delayed responses attributable to the complex environmental conditions.

Subsequent efforts will involve the continuation of channel sounding to gather statistical data on the communication channel within the infrastructures of the fish farm. Additionally, diverse modulations and link layer protocols will be tested to optimize the performance of the network.

The cNODE Modem MiniS is the modem deployed from the North rig and is a commercial off-the-shelf underwater acoustic modem from Kongsberg acting as a benchmark for new and improved communication methods developed for the Observatory. The NORCE software-defined modem (Version 7.37, HiPAP version 4.6.2, μPAP using the same version as HiPAP) shares the same transducer models, which are the components that generate and receive the acoustic signal.

The cNODE Modem MiniS [11] is a device that enables the transmission of data between underwater acoustic transponders or between a transponder and a surface vessel. It is compatible with any HiPAP and μPAP systems [12], which are used for positioning and communication in subsea operations. Acoustic signals use the Cymbal digital protocols, which are QAM direct sequence spread spectrum signals with variable spread factors. These are proprietary and robust methods developed by Kongsberg (Kongsberg ASA, Kongsberg, Norway). The modem has a depth rating of 4000 m, a frequency band of 21–31 kHz, and a data rate of up to 6.0 kbit/s. It can operate with internal or external power, and it has an external connector for configuration and software update. The cNODE MiniS Modem can also perform range measurements between transponders, with an accuracy of 2 cm and a precision of 1 cm. It also has an internal tilt sensor that can measure the orientation of the device in the water column. The modem can be used for various applications, such as positioning of ROVs, towfish tracking, data transfer from subsea sensors and LBL network calibration. It is designed to be easy to use and reliable, with features such as external on/off function, pressure relief valve, battery charger, and configuration software.

The cNODE MiniS Modem has different options for transducers [13]. The selection of transducers depends on the water depth and environment of the application. For the Observatory, the TD30H transducer with a doughnut-shaped beam and TD180 with a hemispherical beam were selected. These should be described with sufficient details to allow others to replicate and build on the published results.

## 7. Data Flow and Presentation

The Internet-of-Underwater-Things (IoUT) was initially decomposed into three software architecture layers: underwater data acquisition (or sensing), network and communication, and data presentation, where the analytics procedures are performed. In more recent studies, an intermediate layer was introduced regarding data management that separates the end-user applications from fusion and processing of data coming from different vendors. This layer also enables the reusability of data. However, this requires the implementation of adequate granularity on sharing mechanisms for automatically identifying and making accessible data that can be shared beyond the initial purpose for which it was collected. In the developments and experiments fostered in the observatory, this layer is also being investigated, bridging data produced by sensor systems operated and managed by different service providers. More especially, an intermediate layer has been deployed at the cloud level in a data platform being implemented.

Concerning the deployment of these software layers into computing infrastructures, the sensing layer is associated with physical sensor platforms. In contrast, the network and communication layers are implemented wherever communication devices are installed to ensure the flow of data. On the other hand, data management can have software components distributed across the different infrastructures in the edge to the cloud computing environments. Lastly, data presentation is typically performed in the end-user software applications computers backed by services provided by cloud infrastructures.

When it comes to the data flow (Figure 7) from the deployed sensors to the end-user applications, data have to transverse heterogeneous communication channels and cross-organizational systems. This heterogeneity introduces serialization and deserialization of data, being presented in different formats according to the system’s needs. These factors affect the interoperability and integrability of the many systems involved in data delivery, and they reinforce the requirement to have quality assurance procedures throughout the delivery pipeline. Data are stored in three facilities by Kongsberg, by Aanderaa (Xylem) and by the Institute of Marine Research (Norwegian Marine Data Facility), and published on the Smart Ocean Data Exploration portal [9].

## 8. Automatic In Situ Control of Sensor Data

Underwater sensors are subject to a harsh environment and must operate in saline water in the presence of debris, of underwater currents, and possibly of noise from nearby installed subsea equipment. A major challenge is fouling on and close to the sensor, particularly biofouling in shallow waters. A smart sensor can monitor its measurement for signs of fouling effects and alert an operator through the acoustic network when an onset of fouling compromising the data quality is detected. In situ quality control of sensor data can thus reduce the costs related to frequent manual inspections of the instruments. Another advantage with locally evaluated data is that the measurement and transmission strategies can be dynamically optimized for battery saving, both on instrument and acoustic transmission nodes. Depending on the intended data use, the sensor may calculate the required information from its raw data and only transmit alarms, trends or averaged data over a desired or dynamically updated time-window. Raw data may be locally stored for a certain period and only transmitted if the sensor receives a request.

In the SFI Smart Ocean project, such sensor self-validation, self-calibration and measurement strategies are investigated, taking into account the acoustic networks restrictions to data transmission. Different approaches to automatic quality control are explored, both data-driven based on machine learning [14] and knowledge-driven based on the combination of oceanographic knowledge of the specific location combined with physics knowledge of the sensor technology [15,16]. The online quality control aims to increase the measurement reliability for long term observations, whereas self-diagnostics and self-calibration also aim to reduce the measurement uncertainty of the deployed sensors [17].

Once data quality is evaluated at the sensor level and transmitted together with the measurement data through the acoustic network, the metadata format and rules for combining metadata from different levels needs to be addressed [18]. This subject is an integral part of the SFI Smart Ocean project and an example of a challenge where the holistic view of the whole measurement and transmission system is a clear advantage.

Another challenge addressed by the project is how to define different degrees of or labels of data quality. Data that are discarded as low quality for one user group can be viewed as sufficient for another. Moreover, what is considered as noise for one user group may be the signal of interest to another. The term data quality must therefore be sufficiently specified and documented in order to increase the data reusability and value for society [19]. For example, if quantifying the annual salinity cycle is the target, then flagging occasional short-lived salinity excursions (which might be either instrumental spikes or rare events related to the passage of sub-tropical water parcels) as bad eliminates noise in the seasonal cycle and represents little loss of fidelity. However, if identification of the occasional presence of small subtropical water parcels which might bring in unusual organisms is the target, it is better not to exclude these results [20]. While, for example, in behavioural ecology, temperature measurements used to determine whether an area is habitable for a particular species usually do not need to be more accurate than one degree Celsius, studies of the effects of climate change-induced heat content changes in the deep sea require uncertainties that do not exceed one hundredth of a degree or even less. Thus, while for the behavioural ecologist, the above-mentioned dataset is of sufficient and thus of “high” quality, for the oceanographer the same data set is of insufficient and therefore of “poor” quality [21,22]. The aspects of data quality have been further elaborated by Nguyen et al. [23].

To ensure the reliability of data and efficiency of data-driven decision-making results, it is important to perform data quality control (DQC) before further employing the data for analytics tasks. This activity can be classified into real-time, near-real-time, or delayed mode depending on the timing that this kind of activity is finished after data collection. Typically, real-time DQC is performed immediately when data are collected. When performed in-situ, this is closely related to the sensor self-validation and self-diagnostics described above. Near-real-time can be performed one week at the latest, and delayed mode can take up to a year to complete [19]. Manual DQC in delayed mode is performed by domain experts, and it complements real-time or near-real-time DQC results to fix potential errors of automatic DQC.

We have developed machine learning models to automate near-real-time DQC for our collected data [20]. To do so, good data will be filtered and then further checked if it contains important information. The filtering process of good data needs to be carried out in the underwater environment, which requires DQC algorithms deployed at sensors. During the data transmission, noise can be generated due to communication issues [23], so DQC also needs to be performed at the cloud.

## 9. Testing of Underwater Acoustic Communication Protocols

A dual-channel acoustic communication protocol has been proposed to enable interoperability, high data rates and networking [24]. The protocol has been specified and developed based on a Subsea Wireless Group context (SWiG) to contribute to standardization efforts in the oil and gas industry but also with the potential to be adopted by other industry sectors. In alignment to Smart Ocean research objectives within underwater acoustic communications across industries, a tight collaboration has been established among research and industry partners, to maximize the use of resources, infrastructure, and joint research outcomes. In this context, the aquaculture research facility offers opportunities to exercise use cases and network scenarios within 120 m water depth, utilizing research hardware, software-defined modems and commercial equipment contributed by the partners where the protocol can be deployed and tested. This will enable proof of concept, early interoperability water testing and performance testing to contribute to a qualification program for the protocol to facilitate further adoption across sectors.

## 10. Discussion

Development of automated sensors, communication systems, and intelligent transfer of data by wireless systems will lead to increased monitoring and understanding of the world´s oceans and coastal areas [20,25] as well as freshwater bodies [26]. Increased use of automated systems will enhance our capability to perform this monitoring, and at the same time cut the associated costs, in comparison with traditional ship-based methods.

Implementing automated sensors and unmanned vehicles for ocean observation potentially reduces the carbon footprint associated with marine research and exploration. Traditional methodologies involved manned research vessels which mainly rely on fossil fuels for propulsion, leading to significant CO_2_ emissions. By switching to automated sensors and unmanned vehicles like underwater drones or AUVs, the need for manned vessels diminishes, resulting in lower fuel consumption and reduced environmental impact. These autonomous technologies can efficiently collect data on ocean conditions, marine life, and climate patterns without the same level of greenhouse gas emissions. Embracing automation in ocean observation not only improves data collection accuracy and efficiency but also aligns with sustainability goals by minimizing carbon footprints in marine research practices.

Will the facility be used? For users there are several factors that are important for choosing to do their testing at such a facility:The area itself is suitable (i.e., depth, sheltering/exposure, etc.);The location is easily accessible (travel distance, cargo loading of transport, etc.);Sufficient infrastructure (access to boat, electricity, data connection, local technical assistance, etc.);Reference system (additional information, benchmark sensors, etc.);Level of security (damage to equipment, open/closed data, etc.).

A major factor determining how useful a test facility is, is its suitability for the needs of the different user groups. Important points to consider are for instance:The possibility to install sites on large depths, for testing instruments under high pressures;The possibility of sheltered or exposed test sites depending on if the user wants to test a prototype under optimal conditions or stress-test equipment in order to reach a higher TRL (Technology Readiness Level).

Another advantage of a shallow and sheltered position is that the boat handling the deployment/recovery can be smaller and the time used will be shorter and you have more days with favourable conditions (i.e., lower cost).

Suitability of the test facility is perhaps the most important factor. If a user wants to test a sensor under high pressure it needs to go to a site with large depths. Alternatively, if rough weather conditions are wanted, the location could be on an exposed open coastline. However, before going to such places the sensors should be qualified for such locations. For less developed and qualified equipment, i.e., with a lower TRL level, a more sheltered and shallow location would be preferable.

The location should be easy to reach; a short distance from the user or from a main harbour/airport/train station. The user does not want to spend too much time on the travel itself. Another factor is to bring the equipment close to where it should be tested. There should be access for a cargo truck or deepwater pier; also space for mobilization of the equipment before deployment or demobilization after retrieval.

Infrastructure services at the site are essential. Most users do not have access to their own ships. There is also a need for fresh water supply for cleaning the equipment, electricity (such as 230 V or 400 V). Already established data connection points ensure a smoother operation. Access to local technicians with practical know-how and local access to necessary tools will dramatically improve both test site operation and user experience. If the test runs over several days, office space and sleeping accommodations will be needed for non-local staff.

When testing a sensor, it is very useful to test it against a reference sensor to conduct a benchmark. Further, when testing in the ocean, even at sheltered locations, you are not in control over all ocean state conditions. Information about other factors that might influence your testing data is therefore very useful. A reference system around the test spot can therefore be of great help when looking for explanations of strange measurements.

Lastly, security at the site is important to prevent your equipment from being damaged. Also, though sharing of data is usually a good thing, there might be times when these should be kept internal to the user. A FAIR data principle states that the data shall be as open as possible, but as close as necessary. This opens for both sharing and sheltering the data, after the need of the sensor owner. For a test site this is an important point to follow.

## 11. Conclusions

The development of a smart underwater observation system based on acoustic communication is a complex task that requires interdisciplinary teamwork. A wide range of disciplines and subject experts are involved in the selection of robust sensors, set up of physics or data-driven quality control on the sensor node, development of protocols for energy-efficient acoustic communication, evaluation of environmental impacts of the acoustic communication, selection of relevant metadata for increasing data usability and reusability, the transmission to the cloud, delayed mode processing, and finally, data presentation and visualization.

## Figures and Tables

**Figure 1 sensors-24-06530-f001:**
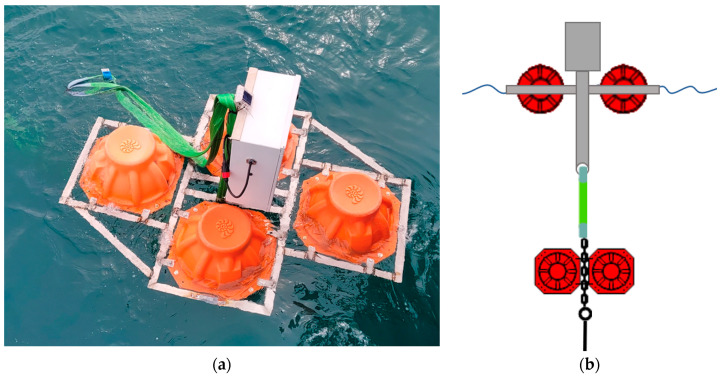
The surface units of the rigs at Austevoll. (**a**) The floating rig, seen from the research vessel. (**b**) Schematic drawing of the surface unit, showing the floaters (in red colour) (four on the surface, two at approx. 10 m depth). The grey unit contains the 4G transmitter and the batteries. The green and black lines are the connections between the units and to the lower parts of the rigs, where the sensors are (shown in Figure 2).

**Figure 2 sensors-24-06530-f002:**
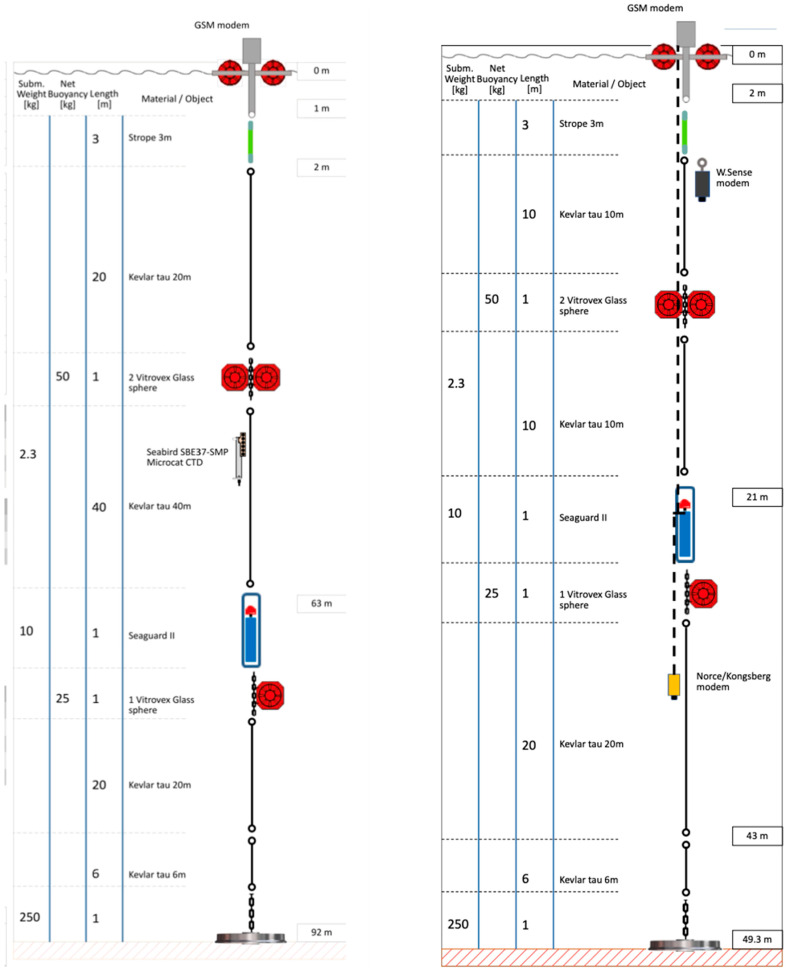
Schematic drawing of the two rigs at Austevoll: South (**left**) and North (**right**).

**Figure 3 sensors-24-06530-f003:**
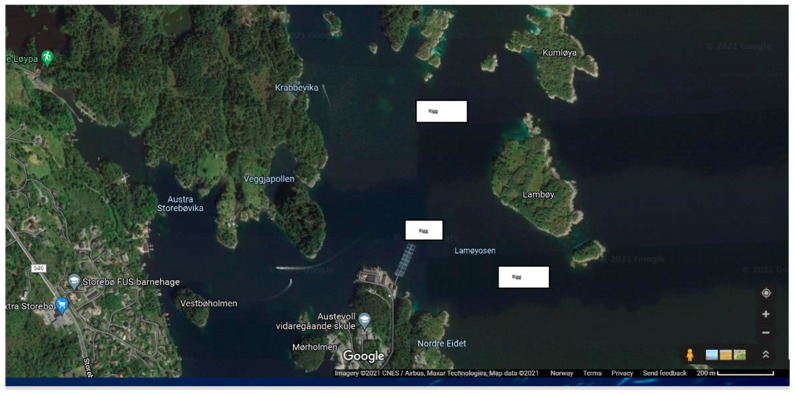
Google maps picture of the facility showing the two sensor rigs and the fish farm facility (centre).

**Figure 4 sensors-24-06530-f004:**
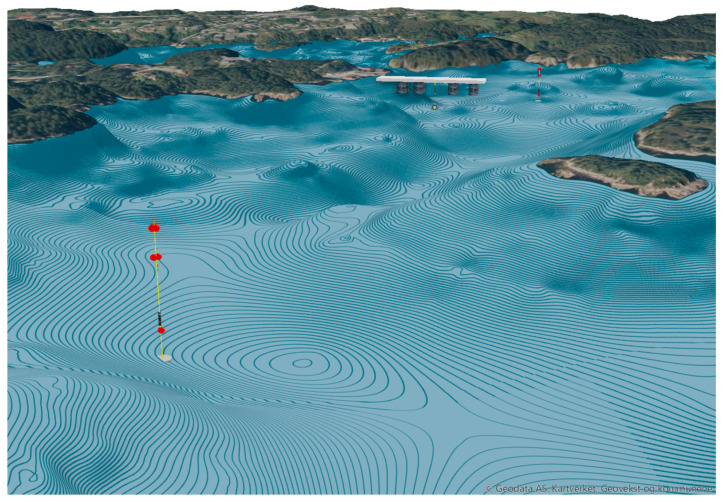
Schematic representation of the bathygraphy of the Austevoll site, with the fish farm facility central and the two sensor rigs North (behind) and South (front).

**Figure 5 sensors-24-06530-f005:**
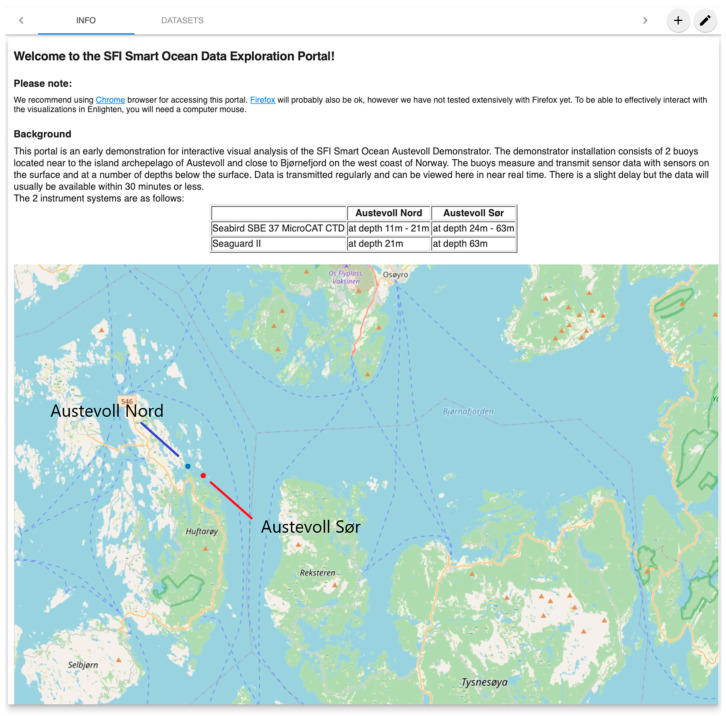
The website of the project SFI Smart Ocean [9]. Data from the rigs and various sensors are published here in near-real-time.

**Figure 6 sensors-24-06530-f006:**
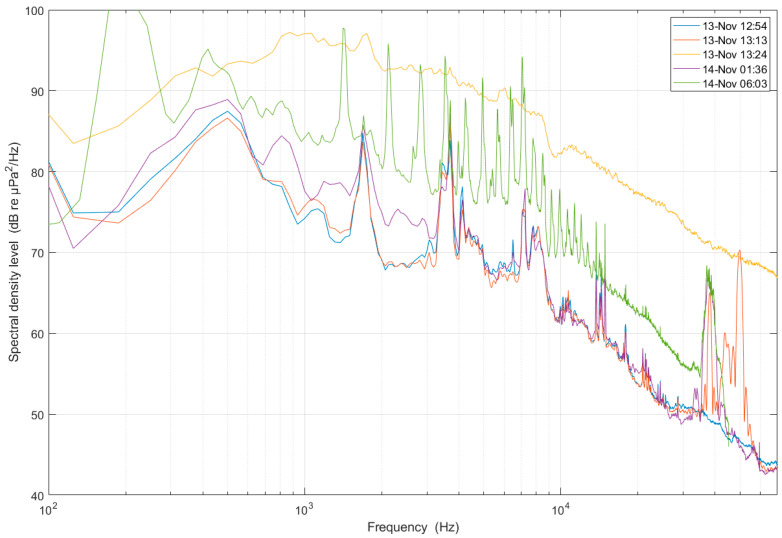
Acoustic noise spectra measured at the Austevoll Research Station.

**Figure 7 sensors-24-06530-f007:**
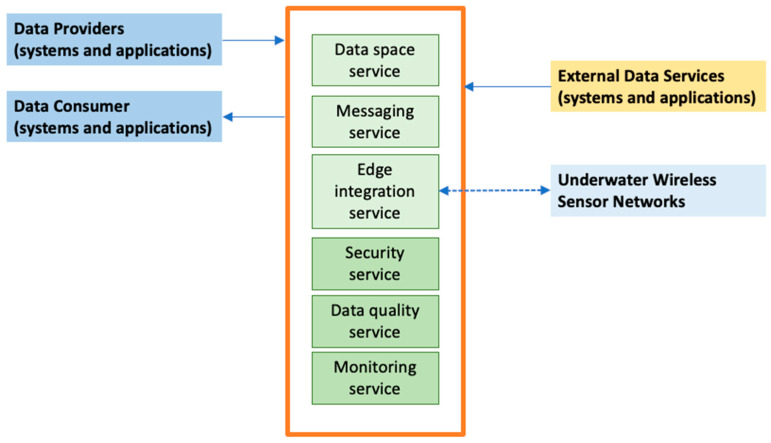
Overview of the software architecture of the SFI Smart Ocean Platform.

## Data Availability

The data are published on websites listed in the manuscript.

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
