# Peer review of "A Modular Smart Ocean Observatory for Development of Sensors, Underwater Communication and Surveillance of Environmental Parameters"

_sensors, 2024, doi:10.3390/s24206530_

Round 1

Reviewer 1 Report

Comments and Suggestions for Authors

The data determination and collection in ocean observation are very important. The authors developed a modular Smart Ocean observatory connected to a large-scale marine aquaculture research facility. The first sensor rigs have been operated for two years. The work will lay a basis for sophisticated use of smart sensors with communication systems in long-term autonomous operation in remote as well as nearshore locations. It is very interesting. I include some specific comments below.

1. The authors claimed that “A major challenge is fouling on and close to the sensor, particularly biofouling in shallow waters. A smart sensor can monitor its measurement for signs of fouling effects and alert an operator through the acoustic network when an onset of fouling compromising the data quality is detected.” How did the smart sensor monitor its measurement for signs of fouling effects? The sensor is usually fouled as time goes on. Would the authors like to explain it in detail?

2. For “Data that is discarded as low quality for one user group can be viewed as sufficient for another. Moreover, what is considered as noise for one user group may be the signal of interest to another.”, would the authors like to give an example?

3. Some related reference should be given, for example, “ These are proprietary and robust methods developed by Kongsberg”.

4. Figure 7 was not showed clear and complete.

Comments on the Quality of English Language

There are some misspelling, such as “traditional ship-based methids” (line 359) and “at a For less” (line 394).

Author Response

The following is the response to reviewer 1. The full reply to both reviwers is in the attached PDF

Reviewer 1

(Comments from the reviewer are in Italic, our answers are in plain Arial typo)

The data determination and collection in ocean observation are very important. The authors developed a modular Smart Ocean observatory connected to a large-scale marine aquaculture research facility. The first sensor rigs have been operated for two years. The work will lay a basis for sophisticated use of smart sensors with communication systems in long-term autonomous operation in remote as well as nearshore locations. It is very interesting. I include some specific comments below.

Answer: We thank the Reviwer for his/her positive comments, and have responded to each of the specific points below.

  1. The authors claimed that “A major challenge is fouling on and close to the sensor, particularly biofouling in shallow waters. A smart sensor can monitor its measurement for signs of fouling effects and alert an operator through the acoustic network when an onset of fouling compromising the data quality is detected.” How did the smart sensor monitor its measurement for signs of fouling effects? The sensor is usually fouled as time goes on. Would the authors like to explain it in detail?

Answer: Thanks for this comment. In this new version, we have referred to the paper by Skålvik et al. (reference No. 16 in the previous version, as well as the the revised version). We hope this reference, and the slight changes in the relevant paragraph of the manuscript, is satisfying.

  1. For “Data that is discarded as low quality for one user group can be viewed as sufficient for another. Moreover, what is considered as noise for one user group may be the signal of interest to another.”, would the authors like to give an example?

Answer Thanks for this comment. In this new version, we have included a new reference to the two more publications, plus one that was already cited in another context: :

1  Jansen P, Shadwick EH and Trull TW (2021). Southern Ocean Time Series (SOTS) Quality Assessment and Control Report Salinity Records Version 1.0. CSIRO, Australia. DOI: 10.26198/rv8y-2q14 

https://data.aodn.org.au/IMOS/DWM/SOTS/reports/CSIRO_SOTS_Salinity_2009-2020_QC_report_ver1.pdf  

Quoting this reference:  «… For example, if quantifying the annual salinity cycle is the target, then flagging occasional short-lived salinity excursions (which might be either instrumental spikes or rare events related to the passage of sub-tropical water parcels) as bad eliminates noise in the seasonal cycle and represents little loss of fidelity. However, if identification of the occasional presence of small subtropical water parcels which might bring in unusual organisms is the target, it is better not to exclude these results.» (reference no 21 in the new version)

2  We have also referred to the the paper by : Waldmann Christoph , Fischer Philipp , Seitz Steffen , Köllner Manuela , Fischer Jens-Georg , Bergenthal Markus , Brix Holger , Weinreben Stefan , Huber Robert (2022). A methodology to uncertainty quantification of essential ocean variables, Frontiers in Marine Science, Vol. 9. DOI: 10.3389/fmars.2022.1002153 (Reference no 22 in the new version)

Quoting this reference:  "While, for example, in behavioralecology, temperature measurements used to determine whether  an area is habitable for a particular species usually do not need to be more accurate than one degree Celsius, studies of the effects of climate change-induced heat content changes in the deep sea require uncertainties that do not exceed one hundredth of a degree or even less. Thus, while for the behavioral ecologist, the above-mentioned dataset is of sufficient and thus of “high” quality, for the oceanographer the same data set is of insufficient and therefore of “poor” quality."

3 We have refered to Nguyen et al. 2023a (reference no. 20  in the original submission as well as the new version), which was also refered to in another context in the original version

We hope that adding this reference, and the associated added sentence in our manuscript, is satisfying.

  1. Some related reference should be given, for example, “ These are proprietary and robust methods developed by Kongsberg”.

Answer: Thanks for this comment, we have added this formulation

  1. Figure 7 was not showed clear and complete.

Answer: Thanks for this comment, which we agree upon. The figure has been omitted from the manuscript, as it did not add value, and the relevant text should be sufficient.

Comments on the Quality of English Language

There are some misspelling, such as “traditional ship-based methids” (line 359) and “at a For less” (line 394).

Answer: Thanks for these comments, which have been taken into account in the revised version.

Reviewer 2 Report

Comments and Suggestions for Authors

Please see my comments in the attached file.

Comments on the Quality of English Language

Minor English correction should be considered for this manuscript. 

Author Response

Responses to reviewers –

Manuscript title: A modular Smart Ocean Observatory for development of sensors, underwater communication and surveillance of environmental parameters in an aquaculture research facility

Manuscript ID: sensors-3035101

Reviewer 1

(Comments from the reviewer are in Italic, our answers are in plain Arial typo)

The data determination and collection in ocean observation are very important. The authors developed a modular Smart Ocean observatory connected to a large-scale marine aquaculture research facility. The first sensor rigs have been operated for two years. The work will lay a basis for sophisticated use of smart sensors with communication systems in long-term autonomous operation in remote as well as nearshore locations. It is very interesting. I include some specific comments below.

Answer: We thank the Reviwer for his/her positive comments, and have responded to each of the specific points below.

  1. The authors claimed that “A major challenge is fouling on and close to the sensor, particularly biofouling in shallow waters. A smart sensor can monitor its measurement for signs of fouling effects and alert an operator through the acoustic network when an onset of fouling compromising the data quality is detected.” How did the smart sensor monitor its measurement for signs of fouling effects? The sensor is usually fouled as time goes on. Would the authors like to explain it in detail?

Answer: Thanks for this comment. In this new version, we have referred to the paper by Skålvik et al. (reference No. 16 in the previous version, as well as the the revised version). We hope this reference, and the slight changes in the relevant paragraph of the manuscript, is satisfying.

  1. For “Data that is discarded as low quality for one user group can be viewed as sufficient for another. Moreover, what is considered as noise for one user group may be the signal of interest to another.”, would the authors like to give an example?

Answer Thanks for this comment. In this new version, we have included a new reference to the two more publications, plus one that was already cited in another context: :

1  Jansen P, Shadwick EH and Trull TW (2021). Southern Ocean Time Series (SOTS) Quality Assessment and Control Report Salinity Records Version 1.0. CSIRO, Australia. DOI: 10.26198/rv8y-2q14 

https://data.aodn.org.au/IMOS/DWM/SOTS/reports/CSIRO_SOTS_Salinity_2009-2020_QC_report_ver1.pdf  

Quoting this reference:  «… For example, if quantifying the annual salinity cycle is the target, then flagging occasional short-lived salinity excursions (which might be either instrumental spikes or rare events related to the passage of sub-tropical water parcels) as bad eliminates noise in the seasonal cycle and represents little loss of fidelity. However, if identification of the occasional presence of small subtropical water parcels which might bring in unusual organisms is the target, it is better not to exclude these results.» (reference no 21 in the new version)

2  We have also referred to the the paper by : Waldmann Christoph , Fischer Philipp , Seitz Steffen , Köllner Manuela , Fischer Jens-Georg , Bergenthal Markus , Brix Holger , Weinreben Stefan , Huber Robert (2022). A methodology to uncertainty quantification of essential ocean variables, Frontiers in Marine Science, Vol. 9. DOI: 10.3389/fmars.2022.1002153 (Reference no 22 in the new version)

Quoting this reference:  "While, for example, in behavioralecology, temperature measurements used to determine whether  an area is habitable for a particular species usually do not need to be more accurate than one degree Celsius, studies of the effects of climate change-induced heat content changes in the deep sea require uncertainties that do not exceed one hundredth of a degree or even less. Thus, while for the behavioral ecologist, the above-mentioned dataset is of sufficient and thus of “high” quality, for the oceanographer the same data set is of insufficient and therefore of “poor” quality."

3 We have refered to Nguyen et al. 2023a (reference no. 20  in the original submission as well as the new version), which was also refered to in another context in the original version

We hope that adding this reference, and the associated added sentence in our manuscript, is satisfying.

  1. Some related reference should be given, for example, “ These are proprietary and robust methods developed by Kongsberg”.

Answer: Thanks for this comment, we have added this formulation

  1. Figure 7 was not showed clear and complete.

Answer: Thanks for this comment, which we agree upon. The figure has been omitted from the manuscript, as it did not add value, and the relevant text should be sufficient.

Comments on the Quality of English Language

There are some misspelling, such as “traditional ship-based methids” (line 359) and “at a For less” (line 394).

Answer: Thanks for these comments, which have been taken into account in the revised version.

Reviewer2

(Comments by the reviewer are given in Italic, whereas our answers are given in plain Arial typo)

Manuscript title: A modular Smart Ocean Observatory for development of sensors, underwater communication and surveillance of environmental parameters in an aquaculture research facility

Manuscript ID: sensors-3035101

The authors developed a system to automatically collect and transfer the data in near real time to the server. In theory, the idea is good and meets the high demand of such system for aquaculture and water quality management. However, the manuscript did not have the basic sections of a standard research, including hypothesis/ research question, results, a clear method... Almost the sections are short and too general to convey the ideas and to help the readers understanding the purposes of the study or the problems will be solved by this study.

The current version of the manuscript is likely a general report on the setting and operation of the system, and it is not really a research paper. Regretfully, I will give a decision of reject the current version of the manuscript but motivate the authors resubmit to the journal with a better version.

Here are my major and minor points that hope to help you: Major points:

  1. Abstract is likely an informative paper of new coming products rather than an abstract of a scientific research. This section should be reconstructed to help the readers with (i) why this module is urgent to develop; (ii) how the experiments were designed? (iii) what are the most important results? And (iv) any outstanding points of this module compared to other products on the market?

Answer: We have not developed a new product intended for the market, for instance the aquaculture industry. Instead, we have developed a test station for sensors, underwater wireless communication, and for treatment of data from such systems, which is generic, and can be used for various test purposes. We have slightly changed the phrasing of the abstract  as well as the Introcuction and hope this has been clearified.

  1. Introduction needs to come with a shortreview of the challenges int is topic and how this paper will solve those problems. The current version of the introduction is too short and too general for the readers to understand the approaches of this study and the motivation to do this study.

Answer: We thank the Reviewer for this comment. The Abstract as well as the Introduction has been changed, and hopefully this is satisfactory.

  1. How is the scale of aquaculture system in this area? Does the system support this sector production or will be used for other purposes?

Answer: We thank the Reviewer for this comment. The Abstract and the Introduction has been changed, and hopefully makes it clear that the intention behind the facility is not to support the aquaculture sector per se, but constitutes a test facility that can be used across marine industries.

  1. Line79-85:you made a copy of these sentences.

Answer: We thank the Reviewer for this comment. This error has now been removed.

  1. Lines 59 -78 are pretty long for only general topics of aquaculture and monitoring while the remaining parts did not mention about the motivation to develop such a new sensors/ IuOT system? Why did you need to develop this? Any novel points in structure, communication, system components compared to other systems? Are there any similar system on the market? How do their performance that motivate you to develop this one?

Answer: We refer to our answers to point 1-3, the changes in the Abstract and Introduction, which hopefully make the motivation behind this study more clear.

  1. The introduction made me feel that this is not a research,rather are reporto f regular work in the industry to build and sell the system to the market. Do you think so? If not, please convince me with suitable ideas.

Answer: We refer to the changes that have been done in the Abstract and Introduction, which hopefully makes it clear that this paper describes a test facility for a wide range of purposes across industries. This is not a system intended to be sold to the market, however, we would welcome collaboration to test new sensors and communication systems within the facility.

  1. Line 94 - 96: why did not you add the parameters of pH, Chl-a and/ or nutrient loading, which are very important for aquaculture system monitoring? Please clarify.

Answer: We thank the Reviewer for this comment. We fully agree that these parameters could have been of very high importance in a system intended to monitor water quality for aquaculture purposes, especially in the case of Low-Trophic Aquaculture. However, as clearified by the rephrased Abstract and Introduction, this facility is not intended for surveillence in any particular industry, but is a test facility for new sensors and communication systems. This is also the underlying point in many of our answers above.It should be mentioned that these parameters are regularly monitored by the aquaculture research station, (i.e. by the Institute of Marine Research), but these data are outside the scope of this particular paper and project.

  1. Fig. 3, any data support or analysis to hep decide the setting positions of the two rigs in the North and South regions?

Answer: We thank the reviwer for this comment. We did not have such data. For a test system, we needed to rigs with some distance betweeen them to provide a facility for testing sensors and communication systems. The positions are arbitrary, and other positions could have been chosen. Indeed, as described in the paper, several experiments are being carried out with instruments in other positions.

  1. Why don’t you have another rig closed to areas around the Austria Storebovika (for instance) to help monitor water quality impacted by fish cages?

Answer: We thank the reviewer for this comment, and refer to the changes in the Introduction and Abstract. Such a rig could indeed be useful for aquaculture puroposes, but would not be necessary in the context and scope of the present paper.

9 .Fig. 1 and Fig. 2 need to come with an explanation of symbols with your colors. It is hard for me to understand the North and South structures seeing the current version of Fig. 2.

Answer: We thank the reviwer for this comment. A better explanation has now been added to the Figure Legend.

10.Line 104 - 113: should be rewrite to enhance English writing

Answer: We thank the reviewer for this comment, and hope the paragraph is now acceptable

  1. Please include a table describing the label/ parameters of the sensors used in the system. You can put the table in the supplementary section.

Answer: We have chosen not to include such a Table. All instruments are covered in the text (underwater communication) or in text as well as Figures (sensors)

12.Line 128: how do you establish a communication using cable? I don’t see it on Fig. 3 or Fig. 4.

Answer: We thank the Reviewer for this comment. Although being intended from the outset, no cable is presently included in the facility, thus the new version does not mention it.

13.Section 8 is too general in describe and explaining for such a system and architecture presented in Fig. 8. As a reader and a user, I would like to know (1) how will the data will be collected and storaged? (ii) how do we validate the integrity and the quality of collected data? (3) how big the data can be in terms of storage and transferring quota? (4) what is the detailed structure of the system? Also, what is the purpose of section 8? It still lack of information even when you want to have a general introduction of data collection, quality check and data transferring.

Answer: These are several highly different questions. (i) has been answered at the end of Section 8 in the new version, (ii) is realated to question 15 and answered there).For (iii) we do not have an appropriate answer, as the amount of data is not a problem for either of the three storage bases. For (iv) we hope the slightly changed section is now satisfactory.

  1. Line 299 - 300: how do we validate the data quality and integrity of the data?

Answer: Here we refer to our extensive answer to Reviewer 1, who also asked about this. We have extended section 9 considerably, and added two more references (20, 21, 22) in addition to our own.  We hope this is an acceptable answer to the question, and also that it improved the quality of the manuscript.

15.How do you protect the monitoring system from wave, wind and saline environment?

Answer: Any instrument in the sea is exposed to the environment. What we can do, is to use robust instruments, and to add advanced procedures to evaluate data quality, which is elaborated in the paragraph above.

16.What is the hypothesis of this study? I did not see any lines in the manuscript.

Answer: Although we thank the Reviewer for this questionThis is a descriptive paper, describing the structure of a complex test facility for sensors and communication systems. As it is purely descriptive, it can not be viewed as a hypothetic-deductive protocol. We hope that the present update of the Abstract and Introduction make this clearer.

17.And where are the results? I did not see any results that support your hypothesis and to lead to the discussion/ conclusion?

Answer: With reference to the answer to question 17, it should be emphasized that the article is of descriptive nature, not hyopthetic-deductive.

18.What are the methods used in this study to build the system/ to solve the problems if any?

Answer: Although we thank the Reviewer for this question, we must sadly add that we do not understand it. The paper is descriptive, and our system can be used for test puropses.

Minor points:

  1. Line 53: typo “e industries. Observations”
  2. Line54:citation needed for“ Observations by  scientific vessels have a long history”
  3. Line 102: “...two at approx.. 10m..” Please not have an incompleted word here or you will need a table of acronyms.
  4. Line 94 - 95: “ and current parameters, temperature, pressure, dissolved oxygen and conductivity”: are temperature, pressure...belong to the “current parameters” or “current parameters” are other parameters? Please carefully check all the sentences like this, if any, in the manuscript.
  5. Line98 need to be back to line97, don’t leave that line alone.
  6. Fig.4,please remove (a)or(b)
  7. Fig. 4, please change color of the contour lines with number of the depths at study site
  8. Fig.2,should have minus signs before the number since you set themeter below 0 m.
  9. Line 166: typo “the cNODE Modem”

10.Line 213, Fig. 7, where are (a) and (b)? 11.Line 225: Double section 7
12.Line 279 should be back to line 278 13. Line 285, in situ should be in situ (italic, author´s comment)

Answer: We thank the Reviewer for all these comments, which were, of course, highly useful. With the exception of point 7, we have responded to all, in a way we hope is acceptable. Regarding point 7, I do not have sowtware to do that now, and the colleague in charge is away on holiday. We therefore kindly ask that this Figure is accepted as is.

Round 2

Reviewer 1 Report

Comments and Suggestions for Authors

The revised manuscript improved obviously, although there are still logical problems.

1. The title should be more concise.

2. There are two parts of 7, i.e. “7. NORCE Software Defined Modem (SDM)” and “7. cNODE Modem MiniS”.

3. The conclusion part should be provided.

Comments on the Quality of English Language

Some expressions should be improved, for example, “and when done in-situ this is closely related to the sensor self-validation and self-diagnostic described above” (line 352) and “ to facilitate further adoption across” (line 378), and “A major factor determining how useful a test facility is, is how suitable it is for the needs of the different user groups.” (line 410).

Author Response

Comment:

The revised manuscript improved obviously, although there are still logical problems.

  1. The title should be more concise.

Response: We thank the Reviewer for this comment. We have shortened the title, and hope it better covers the generic aspects of the study, as well as have deleted the specific reference to the aquaculture research facility.

Comment. 

2. There are two parts of 7, i.e. “7. NORCE Software Defined Modem (SDM)”and “7. cNODE Modem MiniS”.

Response:

We thank the Reviewer for this Comment. These two paragraphs have now been merged into one, hopefully avoiding any misunderstandings. 

Comment:

The conclusion part should be provided.

Response.

We thank the Reviwer for this comment. A brief Conclusion sections has now been added to the manuscript.

Comments on the Quality of English Language

Some expressions should be improved, for example, “and when done in-situ this is closely related to the sensor self-validation and self-diagnostic described above” (line 352) and “ to facilitate further adoption across” (line 378), and “A major factor determining how useful a test facility is, is how suitable it is for the needs of the different user groups.” (line 410).

Response:

Wethank the Reviewer for these comments. We have slightly re-phrased all these sentences, and hopefully they are better now!

Reviewer 2 Report

Comments and Suggestions for Authors

Despite a reject decision from the last review, I checked the revised version of the manuscript as well as the responses from the authors. Unfortunately, the answers did not cover all my concerns and questions, especially the structure of the manuscript and requirements for a standard scientific manuscript. 

Please check other accepted papers on Sensors for examples: 

1. https://www.mdpi.com/1424-8220/24/11/3682

2. https://www.mdpi.com/1424-8220/24/11/3556

3. https://www.mdpi.com/1424-8220/24/13/4168

4. https://www.mdpi.com/1424-8220/24/13/4145

For this reason, I would like to keep my last decision as rejection of the manuscript to be published on Sensor journal. Obviously, this is my personal view point and the editor can find another reviewer or the author may suggest another reviewer to analyze this work. 

Best regards,

Anonymous reviewer

Comments on the Quality of English Language

Minor English editing should be considered for this manuscript. 

Author Response

Comment:

Despite a reject decision from the last review, I checked the revised version of the manuscript as well as the responses from the authors. Unfortunately, the answers did not cover all my concerns and questions, especially the structure of the manuscript and requirements for a standard scientific manuscript. 

Please check other accepted papers on Sensors for examples: 

1. https://www.mdpi.com/1424-8220/24/11/3682

2. https://www.mdpi.com/1424-8220/24/11/3556

3. https://www.mdpi.com/1424-8220/24/13/4168

4. https://www.mdpi.com/1424-8220/24/13/4145

For this reason, I would like to keep my last decision as rejection of the manuscript to be published on Sensor journal. Obviously, this is my personal view point and the editor can find another reviewer or the author may suggest another reviewer to analyze this work. 

Best regards,

Anonymous reviewer

Response:

We thank the Reviewer for his/her response.

We chose to include a reference to one of the suggested papers, as it had significant relevance to our study. This is now included as a new reference in our Discussion.